# Barriers to care for dependent older adults: Brazilian Primary Health Care managers' perspective

Jonas Loiola Gonçalves[1]*, Raimunda Magalhães da Silva[2], Girliani Silva de Sousa[3], Indara Cavalcante Bezerra[4], Christina César Praça Brasil[1], Luiza Jane Eyre de Souza Vieira[2], Fernanda Colares de Borba Netto[2], José Maria Ximenes Guimarães[1], Maria Cecilia de Souza Minayo[5]

1 Graduate Program in Collective Health, State University of Ceará, Fortaleza, Ceará, Brazil, 2 Graduate Program in Collective Health, University of Fortaleza, Fortaleza, Ceará, Brazil, 3 Department of Clinical-Surgical Nursing, Federal University of São Paulo, São Paulo, Brazil, 4 Professional Master's Degree in Health Management, State University of Ceará, Fortaleza, Ceará, Brazil, 5 Foundation Oswaldo Cruz, Rio Janeiro, Brazil

☯ These authors contributed equally to this work.
* jonasloiola10@hotmail.com

**Data Availability Statement:** All relevant data are within the manuscript and its Supporting information files.

## Abstract

This study analyzes the perspective of 16 managers from different Brazilian regions about the barriers to adequate care for dependent older adults in Brazilian PHC. This qualitative approach is anchored in the hermeneutic-dialectic theoretical framework. It builds on the principle that aging has different epidemiological rhythms and multiple specific demands from older adults' biopsychosocial changes. We highlighted several challenges to health services, since family, educational, organizational, and social contexts are fragmented. The 16 managers were selected by convenience from June to September 2019. They interacted individually in a semi-structured interview lasting approximately 30 minutes. The interviews were transcribed in full, and data were organized into four empirically selected themes: barriers due to dependent older adults' family care problems; lack of priority in PHC scheduling; failure in recruiting and training professionals; and lack of a specific public policy. The results indicated weakened families regarding lack of training and overload, especially female caregivers in care routine. Managers recognize the repeated comings and goings and searching for PHC access, highlighting insufficient primary services to organize care. Noteworthy is that high population demand exacerbates inadequate coverage, since lack of professionals with adequate training, priority on the agenda of services, and a public support policy to meet this population's needs fragment healthcare. Thus, it is essential to remedy the barriers recognized by managers by training more experts and qualifying professionals in the service. Society must recognize the stages of aging and support families, especially those most socially needy. The health sector at the Primary Care level has much to contribute to ensuring social protection and healthy longevity for Brazilians.

**Funding:** The authors received no specific funding for this work.

**Competing interests:** The authors have declared that no competing interests exist.

## Introduction

Population aging is almost widespread globally, albeit in heterogeneous epidemiological rhythms. The 2023 Brazilian Census reveals that the elderly population has reached 31.2 million, accounting for 14.7% of individuals, and the growth of this population accounted for 39.8% from 2012 to 2021 [1].

The Pan American Health Organization [2] points out that increased life expectancy concerns managers and formulators of public and social policies in the Americas, traditionally focused on childhood, women's reproductive health, and infectious diseases. The rapid increase of older adults over 80 leads to specific care demands and the need for investments in support and protection policies [3].

In Brazil, population aging generally results from improved living conditions and scientific and technological advances, particularly in public health, medicine, nutrition, and personal and social lifestyle-related care, which deconstructs the idea that this process is linear and homogeneous [4].

Brazil has general and specific policies that benefit older adults' health, with established guidelines, indicators, and targets. Regarding frail older adults, the goal never reached is reducing falls and hospital admissions, which still needs to be improved. The lack of monitoring and assessment of the proposals for this follow-up do not allow us to know the extent of the problems, where competence is lacking and what needs to be implemented or improved. The case of older adults' policies and programs show that more is needed to have a well-formulated proposal: it needs to get off the ground and become effective [5].

For instance, a specific policy like the one in European countries and Canada needs to address the situation of frail older adults. The Ministry of Health created the *Melhor em Casa* ("Better at Home") Program to offer home assistance and care. However, it is designed for any Brazilian Health System (SUS) client and not specifically for older adults. Its logic is based on visits and not on 24-hour care, generally provided by family members [6].

Some relevant local initiatives serve as an example of how to work. However, they are local and restricted and are not a government policy, which is the case of the *Maior Cuidado* ("Greater Care") Program of Belo Horizonte, Brazil, which offers formal caregivers for socially vulnerable dependent older adults without family caregivers. São Paulo has the *Programa Acompanhamente de Idosos* (PAI) ("Elderly Monitoring Program"), which provides a multidisciplinary staff to support the most fragile and dependent, including meeting their needs of daily living. In Rio de Janeiro, primary care offers care to bedridden older adults. However, health workers and the Primary Health Care (PHC) staff recognize they can do very little and cannot assist all older adults in need [6].

In the case of Brazil and other international settings, PHC's attributions (non-selective) are promoting health, preventing diseases, and providing care [7]. However, Yagi *et al* [8] commented that, in the case of older adults, care needs to be more cohesive, sufficient, and discontinued, but it does not ensure comprehensiveness and adequate attention to this audience's needs. More often than not, professionals offer misguided guidelines that conflict with patients and family caregivers.

It is important to note that, in the Brazilian situation, before the promulgation of the Federal Constitution, the elderly population suffered from incipient and charitable government actions. From its institution, the SUS was created, a health instance focused on the universal right to health. An articulation with the Unified Social Assistance System (SUAS) emerged in the search to overcome difficulties and guarantee social protection for the population based on support for people, families and their communities [4, 9–12].

Therefore, structuring an Elderly Care Network regulated by PHC requires close communication between the support systems for older adults; however, in Brazil, there are still persistent organizational, financial, structural, and social barriers that directly imply articulation of care as recommended by other countries. Care planning should be individualized and needs to consider risk stratification based on an assessment with pre-established criteria and indicators [9–12]. The aspects that guide and underpin the Brazilian PHC and Elderly Home Care management still need to deepen knowledge about aging and create strategies and practical instruments for action [4].

We can find the most systematized experiences from the point of view of management and provision of human and social conditions to care for dependent older adults and their caregivers today in European countries, although all are criticized when assessing how care occurs in practice. First, longevity is not officially seen as a problem but a challenging social event for different societies. Their established policies and strategies are premised on integrating social and health services at the primary, secondary, and tertiary levels [6].

Countries such as Denmark, Germany, France, and Spain take over the role of the State in taking charge of the frailest, organizing services to cause the least impact on the lives of those who care for them. They offer help at home so caregivers can continue working. The Canadian experience includes activism of caregiver associations that claim tax exemption, burden relief, and reduced working hours for those absent to care for a dependent family member. Although Canada has a national policy on the subject, each state applies its benefit and care strategies differently [13].

In Brazil, laws and protection policies are directed to the Organic Law of Social Assistance [14], the Brazilian National Policy for Older Adults [15], the Statute of Older Adults [16], the Brazilian National Primary Care Policy [7], the Brazilian National Health Policy for Older Adults [17], the Brazilian National Commitment to Active Aging [18], and the Guidelines to Care For Older Adults in the SUS [19]. However, these legal devices focus on something other than dependent older adults, making the experience of longevity in the country increasingly challenging [4].

Thus, based on an assumption that care for frail older adults requires continuous care, the question developed in this article emerged as follows: what are the barriers to providing adequate care for dependent older adults in PHC?

## Objective

The study sought to analyze managers' perception about barriers to care for dependent older adults in Brazilian PHC.

## Materials and methods

The empirical material of this study comes from a multicenter project entitled "*Situational study on dependent older adults living with their families, aiming at subsidizing a care and support policy for caregivers*", developed by a group of researchers linked to Brazilian public and private universities. The study is guided by triangulation assumptions, considering the participation of places, people, researchers, theories and intramethod [20]. It involves the understanding of PHC's work (health workers, professionals who work in units and managers) in the face of the interlocutions of the situation of older adults, their families and family caregivers [6].

We developed a qualitative study anchored in the hermeneutic theoretical framework that is based on the understanding of social processes, communication of daily living, common

sense, considering that human beings are permeated by their historicity, and interconnections come from social and cultural contexts [20, 21].

Furthermore, we tried to analyze a consensus of the experiences, senses, meanings and symbols contained in participants' statements. Thus, a comprehensive reflection emerges on the barriers pointed out by PHC health managers regarding the adequate provision of healthcare to dependent older adults living at home [20, 21].

The study was conducted in Araranguá, Santa Catarina (SC), Brasília, Federal District (FD), Fortaleza, Ceará (CE), Manaus, Amazonas (AM), Porto Alegre, Rio Grande do Sul (RS), Belo Horizonte, Minas Gerais (MG), Rio de Janeiro, Rio de Janeiro (RJ) and Teresina, Piauí (PI). These municipal units were selected by convenience, and only Araranguá is not a state capital. The municipal units were selected because all of them (including Araranguá) are hub regions for meeting the SUS demands, providing users with the necessary equipment and resources for care in primary, secondary, tertiary, and ultra-specialized care areas [20–23].

Study participants form a sample of 16 managers from five different country regions, distributed in: Araranguá, SC (4); Brasília, FD (1); Fortaleza, CE (2); Manaus, AM (2); Porto Alegre, RS (1); Belo Horizonte, MG (4); Rio de Janeiro, RJ (1); and Teresina, PI (1). The number of participants varied between the municipalities because of difficulties scheduling interviews and assigning multiple activities to the local management.

Selected by intentionality and because they are firmly integrated with the local management and with responsibilities inherent in managing care for dependent older adults, we have managers who coordinate and are responsible for managing PHC units, programs and specialized services offering healthcare to dependent older adults. The researchers sent an invitation letter containing the objective and a summary of the research assumptions to the Municipal Health Departments. Specific invitations were sent to managers working with older adults' health and Basic Health Units (BHU) coordinators (Fig 1) [22].

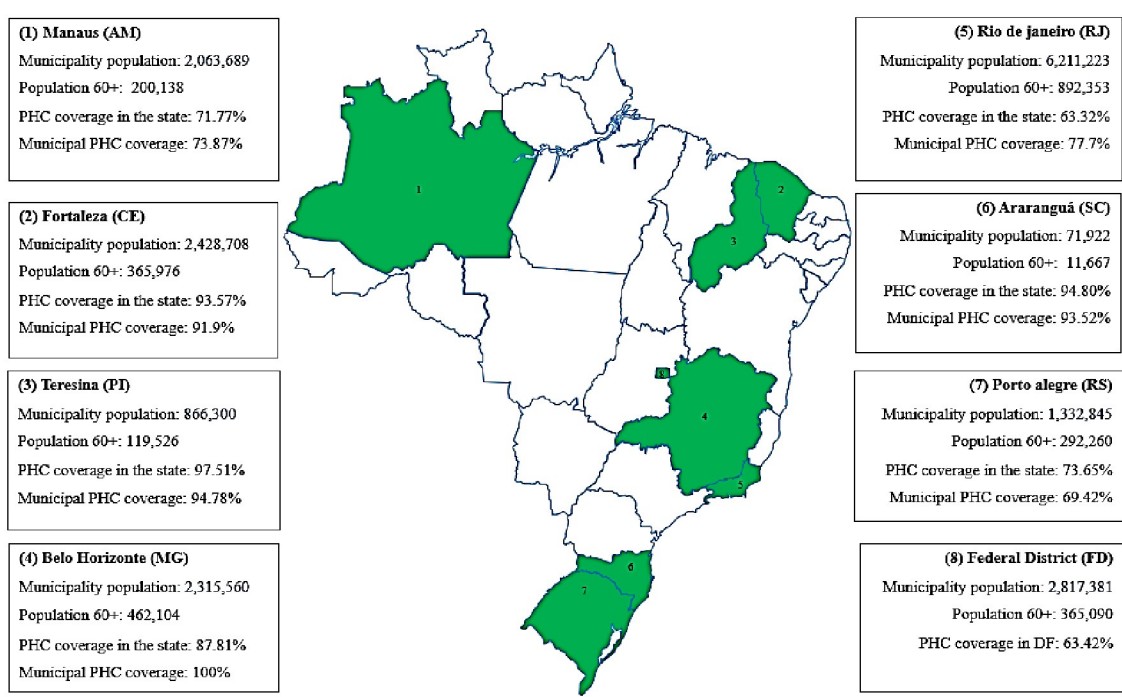

**Fig 1. Characterization of the municipalities participating in the study according to population rates and Primary Health Care coverage.** Brazil, 2019.

The selection for interviews focused on those who, in the care organization chart, were responsible for older adults' health at the municipal level, on PHC Units managers and specific programs for older adults, and those who had at least six months of health management experience. Those absent from their work activity due to vacation, leave due to health problems, or maternity leave were excluded.

Immersion in the field and data collection took place from June 1 to September 30, 2019, simultaneously, in the five regions surveyed, and was the subject of a wide range of issues. The main ones were meeting with older adults and their families, observing their living, health, and dependency conditions, and meeting with healthcare professionals from these places. We presented a preview of results in a collection registered in a thematic issue of the *Ciência & Saúde Coletiva* journal [23]. The managers, the main stakeholders in this study, were heard through a semi-structured interview at a pre-established date, time, and place.

Data collection was supported by a semi-structured interview prepared by consensus by a team of researchers, who had degrees in nursing, medicine, physiotherapy and psychology, of both gender, with previous training on the instrument. Dialogue with managers started with a triggering question about the barriers they observe to properly manage care for dependent older adults in PHC [24]. The interviews were conducted individually, in a reserved room chosen by the interviewees, lasting at least 30 minutes and maximum of 90 minutes. There were no refusals on any withdrawals to answer the questions.

Statements were fully transcribed and organized manually, without software support. The analysis followed interview transcription, material reading and rereading, data organization, classification of significant categories, and interpretation [20]. After discussion of theoretical saturation and data grouping, the following categories were consolidated: barriers due to dependent older adults' family care problems; lack of priority in Primary Health Care scheduling; failure in recruiting and training professionals; and lack of a specific public policy.

This study followed the Consolidated criteria for Reporting Qualitative research (COREQ) [25]. It was submitted and approved by the *Fundação Oswaldo Cruz*/Rio de Janeiro Research Ethics Committee, through the Certificate of Presentation of Ethical Review (44615315.0.0000.5240) 1.326.631, under Resolution 510 of April 7, 2016 [26], which regulates human research with methods anchored in human and social sciences.

Participants confirmed in writing their agreement through the Informed Consent Form, signing two copies, one for the researchers to file and the other delivered to participants. The results are coded with Word Manager, the interview number, and the place where the activity was held to preserve secrecy and anonymity.

## Results

Comprehensive inferences result from the relevance defined by managers who emphasized the following topics: barriers due to dependent older adults' family care problems; lack of priority in PHC scheduling; failure in recruiting and training professionals; and lack of a specific public policy.

### Barriers due to dependent older adults' family care problems

Managers stated that family structure and dynamics is one of the barriers that compromise care for dependent older adults, as it involves body hygiene, attention and care from family members, regardless of whether families are nuclear or uninuclear.

They highlighted the importance of intersectoral and articulated actions so that care is made viable, reinforcing that the family base is the pillar that unfolds and anchors the various

dimensions of care that emanate from older adults. However, they do not mention the family structures and dynamics created in relation to the Brazilian dimension.

In general, managers pointed out that several factors exceed institutional possibilities as they exist today for consolidating care for dependent older adults. The first criticizes the family and its difficulties in welcoming frail older adults and implementing a care plan.

> You can have ten children who will pass the responsibility on to another, and no one will want to take care of them. We are often forced to call the social service to pressure the family to provide care, saying it is an obligation. If you don't have a bath, a clean house, affection, and a distraction, this impacts society and management, and will demand a little effort from everyone.
>
> (Manager 1—Araranguá)
>
> We understand that all the problems are at the family level.
>
> (Manager 1—Manaus)
>
> Most cases that come up for social assistance are family negligence.
>
> (Manager 2—Belo Horizonte)

Faced with barriers in the family setting, managers struggle to plan care provision because professionals' prescriptions need to be followed by someone who can understand and maintain them. Difficulties arise even in maintaining specific programs.

Moreover, planning care provision becomes complex, due to the absence of a family caregiver to take over, hold accountable and maintain this care. In this regard, gaps are identified about the course of aging and limitations inherent to this process.

> Their children grew up. They left, and the older adults lived alone without any care. So, our difficulty is providing reasonable assistance to this older adult without them having a caregiver
>
> (Manager 1—Rio de Janeiro/RJ).
>
> There is great neglect by family members. It may be due to a lack of guidance, information, or something else. I know some situations could be prevented
>
> (Manager 2—Manaus).

One of the managers recognizes that care needs go beyond the actions performed at home, whether by Home Care Service or family caregiver, such as loneliness, relational and affective voids.

> There is no family. She cannot commute, so she stays at home. We have some older adults in the Home Care Service. She doesn't get better because the family doesn't let her date or go dancing. She needs a partner. Her loneliness is not about family but emotional
>
> (Manager 2—Araranguá).
>
> So, you know, sometimes, there is a limit to the program. As long as he has his family and the services there, the family, the program continues, but an older adult who does not have a family caregiver, a back-up, then it is impossible for us to maintain the "Greater Care" Program
>
> (Manager 2—Belo Horizonte).

Participants point out that the family's unpreparedness to address older adults' specific situations directly affects care management. However, they add that there needs to be more financial resources and State support for older adults. This omission leads to fragmenting, disrupting, and upkeeping good practices.

> The family member is unprepared to take care of this older adult. We realized the great difficulty of the family. They get here sometimes needing help.
>
> (Manager 4—Araranguá)

> The weakest side is that of the family because it is the one that has no resources. It's what is criminalized when missing. You criminalize the family that neglects, takes care of badly, or mistreats, but you don't verify under what circumstances that family is being forced to take care of. The State is sovereign there, just watching.
>
> (Manager 2—Belo Horizonte)

> Families are often unable to take the older adult to the health unit. . . which makes it very difficult.
>
> (Manager 1—Fortaleza)

The statements mentioned above reveal different nuances: in general, managers complain about family members' negligence and unpreparedness. However, one of them highlights the difficulties of the poorest people in providing care, adequate food, and medication for older adults. They should not be held accountable for negligence.

> Family members say they will help, but they only take the retirement benefit this older adult receives, but they do not provide care!
>
> (Manager 3—Belo Horizonte)

> They think that, at home, they are useless older adults. It is often because they have a grandson at home or a family member who does not have a conversation they wanted to have.
>
> (Manager 1—Manaus)

> Many of these older adults may need a family or a robust social circle. They need the government to be concerned and care for them, especially on medication time, food, and bathing, because they are there alone.
>
> (Manager 2—Belo Horizonte)

Today, we observed a paradox experienced by families with older adults in their homes. Soon these are usually ageism victims. From non-recognition of older adults as essential people in the family circle or by society, the burden and unpreparedness of those caring for them and their impoverishment emerge, along with lack of financial conditions to pay for formal caregivers.

> If caregivers were better prepared, they would sometimes be successful. However, they need to be more prepared, without direction and a little left aside. They are also tired and repeat, "I can't".
>
> (Manager 3—Araranguá)

Older adults do not just need to be fed and bathed but also receive attention and affection. Many family members who are caregivers do not have patience, as they are sick too, because it is complicated.

(Manager 2—Manaus).

**Lack of priority in Primary Health Care scheduling.**   Care for dependent older adults, their intersectoral and intrasectoral management challenge compliance with the inseparability between care and health management is a Brazilian health system guideline. Older adults repeat comings and goings in the PHC service, acknowledging that the health service still "does not have a leg for everything". Demands about aging emerge as a problem, but managers try to guarantee the right to health, even with situations that weaken healthcare for older adults in Brazil.

Federalism stands out in the Brazilian reality, composed of three federative entities, and is not easily comparable due to its particularities, making PHC implementation a particular challenge in Brazil. Disarticulation of PHC and Family Health Strategy in care for dependent older adults suffers from barriers to implementing universal primary services.

Public health cannot reach all older adults. These patients keep repeating their comings and goings in PHC, repeating. . . because public health does not have the means for everything.

(Manager 2—Araranguá)

This "Greater Care" Program offers caregivers for older adults who are in areas of the Social Assistance Reference Center (CRAS) and high social vulnerability. However, it is reduction after reduction.

(Manager 4—Belo Horizonte).

Managers point out that the insufficient Health Care Network is an obstacle, particularly in care and home care. Participants understand that this need is more acute in failures in welcoming and follow-up due to the following:

Regarding scope and welcoming, care for older adults still leaves much to be desired.

(Manager 2—Belo Horizonte)

It's a big problem, and it will increase a lot. I want to see later the problem of home confinement, the bedridden person, the fragility of social relationships and the massive demand on the Public Prosecutor's Office.

(Manager 1—Teresina)

The current Family Health strategy can only meet some of the demands. It is not just primary care but specialized care as well.

(Manager 3—Araranguá)

Managers point out that older adults' health/illness demands expose the asymmetrical relationship between an insufficient number of professionals and the need to cover these families with quality care and under the coordination of PHC. This disconnection hinders the provision of person-centered care, social protection and, above all, the universality of healthcare at the primary level.

The challenges are to meet this growing population in the city and demand shelter, care and protection.

(Manager 4—Belo Horizonte)

We have a massive demand from older adults. This demand is huge here in Teresina.

(Manager 1—Teresina)

Our units are overloaded. We have several inhabitants well above what the Brazilian National Primary Care Policy (PNAB) recommends for each strategy. So, this hinders providing more detailed care for older adults.

(Manager 4—Araranguá)

We have, in the city of Belo Horizonte, the "Greater Care" Program, which is social assistance, and is giving results, but it needs to be expanded. We increasingly see the need for greater care for the population and these caregivers in the low-income population, who really have no one to care for (Manager 1—Belo Horizonte).

Considering the high population demand for PHC and home care, managers are aware of the health staff overload, who are in charge of fulfilling priority agendas. They emphasize that the staff tries to do their best in the face of the care demand, counting on the participation of third sector institutions, but they are not enough to meet these demands that have been repressed for a long time.

The staff already supports other things. So, on this health issue, the Family Protection and Comprehensive Care Service (PAIF) staff is very overloaded. This situation brings uncertainty about how to proceed with this professional.

(Manager 3—Belo Horizonte)

This is one of the most significant difficulties for municipalities today. Strategies for work bottlenecks and professional overload.

(Manager 3—Araranguá)

The staff tries their best, but we need more support. We have partner institutions, but more is needed. Our population covered by the Family Health Strategy is an extremely old population.

(Manager 1—Rio de Janeiro).

Concretely, this situation results in frail older adults not receiving the necessary care and treatment continuity. Managers showed they were aware of the problems. Several of them are very sensitive to older adults' suffering and seek to improve the quality of services, usually without great success.

There was a time when I had nothing else to do in relation to professionals. The department is unable to absorb all patients for therapy with a psychologist. Social assistance is unable to visit everyone.

(Manager 1—Araranguá)

There is no way for primary care to achieve this. Visits every fortnight. We have primary care and we advise. So, what happens? They call me a lot, and I call the nurse, "Look,

can you go there, give that person some attention?". So, we adapt to the incoming demand.

(Manager 4—Araranguá)

Work processes to provide greater resolution are seen by several managers as an attempt to put out the fire as they focus on solving specific situations with questionable quality:

Professionals only try to put out the fire and do what is agreed. So, we go there and do it and say we did it.

(Manager 3—Araranguá)

We are concerned with immediate action. Professionals are suffocated in solving problems. They want to know about something other than monitoring. They want to solve patient problems. They need to think about quality.

(Manager 1—Teresina)

**Failure in recruiting and training professionals and lack of a specific public policy.** Managers point out that developing skills and competencies for caring for older adults' health needs to be more specific, targeted, and minimal. Educational failures reverberate in PHC and home care services.

We work with a staff who needs to obtain the minimum skills to provide this care to the audience.

(Manager 1—Fortaleza)

Along with the complex care process, turnover in health services has become a widespread problem in the Brazilian health sector after the rules for outsourcing Primary Care services. These precarious work ties and the shortage of personnel reduced the care capacity. There is no guarantee of continuity, and the flow of the treatment process and access to adequate care has been broken.

The change of professionals, nurses, and doctors with their profiles and role needs more articulation.

(Manager 4—Araranguá)

When I arrived here, we had a social worker at the unit. Then, we started losing specialists. There was a social worker who helped us a lot. Many servants got scared and asked to leave with the expanded Family Health.

(Manager 1—Rio de Janeiro)

In the daily life of the manager who works in health, there need to be more resources for older adults, and the main thing needed is human resources.

(Manager 2—Fortaleza)

The "*Mais Médicos*" (More Doctors) Program contract for doctors is over. We only had one doctor in the unit, so it is challenging. The professionals recruited are for the service

provided. They come and go, both doctors and nurses. Employment relationships are fragile.

(Manager 1—Fortaleza).

The lack of a priority agenda and planning focused on older adults' health in PHC and home care is seen by managers as a negligence factor in the daily practice of services.

There are no goals to be achieved for older adults, as in the case of children. I have yet to set a goal to say how many appointments an older adult should have.

(Manager 1—Fortaleza)

The service needs to plan care. This situation needs to be reversed because we are still very reactive and waiting for the incoming demand.

(Manager 2—Belo Horizonte)

Faced with several barriers to these two essential services, managers consider that the problem begins in Brazilian public and private universities, where there needs to be more interest in training to take care of older adults.

We left the medical and nursing colleges thinking about prenatal care, childcare, tuberculosis, leprosy, immunization, hypertension and diabetes, so we fragmented.

(Manager 1—Rio de Janeiro)

We received interns from several undergraduate courses. Only some bring initiatives geared toward older adults. Most are for women and children. The municipality itself complies with this.

(Manager 1—Fortaleza).

Not only training is questioned by managers, but mainly the lack of a public policy to strengthen care for older adults.

What I demand from senior managers are policies aimed at older adults. Most target children, women of childbearing age, and little is said about older adults.

(Manager 1—Brasília)

Most dependent older adults result from poor monitoring of hypertension and diabetes. In other words, the health system does not have an adequate care model for monitoring chronic conditions.

(Manager 2—Fortaleza)

In relation to care, the big issue is often politics today within the city of Belo Horizonte, because we see the needs becoming more and more present

(Manager 1—Belo Horizonte).

## Discussion

Given the barriers pointed out in the study, managers' considerations, according to which the provision of care to older adults occurs in a scenario of "attempting to put out the fire" in both PHC and home care, are uniform. As only the family takes over or neglects care for longevous people, unsurprisingly, the main criticisms regarding unpreparedness, mistreatment and negligence are directed at them.

A paradox is currently experienced by families with older adults in their homes. More than 90.1% of these people help maintain or provide for families alone [27]. The greatest problem occurs when older adults lose autonomy and need someone to care for them. In a study on dependent older adults, of which this article is part, we observed that most women are the caregivers, and the family delegates all responsibility to them [28]. However, some older adults do not have family members or are alone.

The family barriers that permeate the accelerated growth of the older population contradict the low birth rates as well as the changes in marriage rates and the insertion of women in the labor market. It is evident that Brazilian women are increasingly taking over the role of provider, but still maintain responsibility for caring for those who are dependent, changes that directly affect families' ability to guarantee and offer care to older adults [27].

Furthermore, these family caregivers' work is often uninterrupted and solitary, without the support of health services and public protection policies. There is a need for public support policies for dependent older adults and their caregivers, with the aim of overcoming barriers and subsidizing better care for everyone [27, 28].

Although the subject is only addressed at the end of their speeches, managers mentioned the need for greater priority in the Brazilian health system for older adults, where a "blackout" of care arises. The calls for better care in the face of the exponential increase in the number of older adults as well as their participation in society are inconceivable [29].

Although perhaps managers seem unaware, there is a Brazilian National Health Policy for Older Adults, whose established objectives are "to recover, preserve, and promote their autonomy and independence, to direct collective and individual health measures for this purpose, aligned with SUS principles and guidelines. All Brazilian citizens aged 60 or older benefit from this policy". The Ministry of Health ratified this policy through Ordinance 2.528 of October 2006 [17]. The ordinance guidelines include investment in healthy aging, comprehensive and intersectoral care, welcoming priority, provision of resources to assist this population, professional training, the need to offer information to the audience, and international exchanges to improve services.

The text that presents the policy and other guiding documents [17–19] includes a set of manager attributions. The municipal attributions are as follows: (1) developing technical standards for care services; (2) defining financial resources for this purpose; (3) discussing and elaborating goals to be achieved; (4) promoting care intersectoriality; (5) establishing training mechanisms on the subject for professionals; (6) establishing monitoring and evaluation indicators for this policy; (7) presenting and agreeing on an action plan at the Municipal Health Council.

The policy document [17] mentions dependent older adults in two stages, in the introduction and in the chapter that addresses action comprehensiveness, highlighting a concern when managers are unaware of political issues, especially with regard to the issue of dependent older adults. In the 2013 Brazilian National Health Survey, they found 8.4% of older adults dependent on others to perform activities of daily living and 22.0% to perform instrumental activities of daily living, showing that the percentage of older adults who are losing autonomy increases while the number of those aged 80 and older is growing rapidly [30].

Therefore, given the data and the empirical reality shown in the text, we highlighted the need to follow the path of more socially developed countries, promoting a specific policy for dependent older adults, including managers, professionals, family members, and Brazilian society, rapidly advancing towards longevity. Those working in PHC need to master the issue of aging and the Brazilian National Policy Guidelines, in addition to establishing indicators and goals related to care for this social segment. Adequate and systematic care for those who are healthy or who only need exercise, good nutrition, and medication for continuous use reduces the number of those who cross the dependence threshold.

Portuguese and Thai researchers point out that different care models are necessary to ensure care centered on older adults, and there is a need to reach out to family members/caregivers [11, 31, 32]. These authors highlight as priority needs continuing education strengthening for professionals and better working conditions to prevent turnover, given the observed lack of care for older adults in all regions of the country.

However, a service open to the general public, such as PHC units and Community Health Workers, cannot respond to dependents who need help 24 hours a day. This follow-up of older adults who have lost their social, mental, and physical autonomy requires a program that encompasses and values their family caregivers. It establishes needs and care levels (medical, psychological, physiotherapy, social) specific to each case. Health and social assistance professionals must jointly implement specific equipment and services. Two crucial points for this group are offering adequate and continuous training and supervision focusing on care continuity.

When studying the effectiveness of training caregivers for home care and comparing it with a control group without training, Ugur [33] highlights that those who received training showed positive results for sick older adults. Care continuity requires systematic planning so that bonds are not broken, which requires a trained staff to avoid failures when replacement is necessary [31]. Caregivers also need care geared to their mental and physical health, so they do not fall sick [34].

While this specific policy is not forthcoming, special attention needs to be given to family caregivers. A study conducted by Brazilian researchers points out that multiple inequalities, work overload, the multimorbidity of the people they care for, and their impoverishment are a reality that is experienced very strongly [6], which is reinforced by the World Health Organization document called "Decade of healthy ageing: baseline report" [35].

Finally, many societies are still seeking care standards for dependent older adults. A study [36] with several Asian countries, including Japan, Korea, Thailand, China, Indonesia, and the Philippines, points out that, except for the Philippines, some political decisions regarding health service reforms prioritize older adults in all other countries. Thailand and China promote similar actions in favor of population longevity. China advances health and social assistance, and Thailand focuses on home care.

It is a consensus among the authors that a health system centered on the hospital and the medical professional is not sufficient to meet dependent older adults' demands, as there is a need for training, public policies, and dialogue with society. Investments in primary care to reduce and consolidate care are necessary in a society of older adults whose future is already present, including in low- and middle-income countries. The authors highlight that Japan has the most consolidated primary care services [36].

The Japanese government implemented the "Community-Based Integrated Care System" to offer practical support to people at the end of life, guided by four elements: self-help, mutual help, social solidarity care and government care [36]. Japan's organizational model entrusts the implementation of a specific national policy for dependent older adults to the municipalities. It gives substantial autonomy to the regions but with common guidelines for the whole country. These and many other examples exist and can be adapted [36, 37].

An example that could be adapted to the Brazilian reality, based long-term care implementation, concerns institutional investments that meet health managers' needs for the proper consolidation of PHC and the family health strategy, thus strengthening intersectoral relationships centered on home care. Furthermore, it is necessary to take better care of older adults in Brazil and the world that is aging [36, 37].

The limitations of this study include the unavailability of carrying out focus groups between managers and Federative Units. Another limitation is highlighted by access to data collection, given the time that management has for the interview, limited physical spaces and contexts of urban violence that directly implicate research processes in Brazil.

## Conclusions

The barriers faced by managers for caring for dependent and monitored older adults in the Brazilian PHC are related to the family, service organizational conditions, lack of training and professional competence, accompanied by political, social and financial obstacles both of the families and the governmental/organizational spheres. Thus, the care offered generates barriers for the family, caregivers, professionals, managers, and older adults who live alone and with greater dependence.

Therefore, there is a need for more professionals, adequate financing, and health policy consolidation for both primary care and older adults so that services can offer comprehensive care in the face of increasingly high demands. Finally, for the good of society and families, it is necessary to create a specific policy suitable for people with social, physical and mental dependencies, in addition to providing PHC with knowledge, strategies and instruments to assist older adults. The health, social assistance and human rights sectors are called to this noble mission in favor of those who helped build the country!

## Supporting information

**S1 File.**
(DOCX)

## Author Contributions

**Conceptualization:** Jonas Loiola Gonçalves, Raimunda Magalhães da Silva, Girliani Silva de Sousa, Indara Cavalcante Bezerra, Christina César Praça Brasil, Maria Cecilia de Souza Minayo.

**Data curation:** Jonas Loiola Gonçalves, Raimunda Magalhães da Silva, Girliani Silva de Sousa, Indara Cavalcante Bezerra, Christina César Praça Brasil, Luiza Jane Eyre de Souza Vieira, Maria Cecilia de Souza Minayo.

**Formal analysis:** Jonas Loiola Gonçalves, Raimunda Magalhães da Silva, Girliani Silva de Sousa, Indara Cavalcante Bezerra, Christina César Praça Brasil, Luiza Jane Eyre de Souza Vieira, Fernanda Colares de Borba Netto, José Maria Ximenes Guimarães, Maria Cecilia de Souza Minayo.

**Funding acquisition:** Raimunda Magalhães da Silva, Girliani Silva de Sousa.

**Investigation:** Jonas Loiola Gonçalves, Raimunda Magalhães da Silva, Girliani Silva de Sousa, Indara Cavalcante Bezerra, Christina César Praça Brasil, Luiza Jane Eyre de Souza Vieira, Maria Cecilia de Souza Minayo.

**Methodology:** Jonas Loiola Gonçalves, Raimunda Magalhães da Silva, Girliani Silva de Sousa, Indara Cavalcante Bezerra, Christina César Praça Brasil, Luiza Jane Eyre de Souza Vieira, Fernanda Colares de Borba Netto, José Maria Ximenes Guimarães, Maria Cecilia de Souza Minayo.

**Project administration:** Jonas Loiola Gonçalves, Raimunda Magalhães da Silva, Girliani Silva de Sousa, Indara Cavalcante Bezerra, Maria Cecilia de Souza Minayo.

**Resources:** Jonas Loiola Gonçalves, Raimunda Magalhães da Silva, Girliani Silva de Sousa, Indara Cavalcante Bezerra, Christina César Praça Brasil, Luiza Jane Eyre de Souza Vieira, Maria Cecilia de Souza Minayo.

**Software:** Jonas Loiola Gonçalves.

**Writing – original draft:** Jonas Loiola Gonçalves, Raimunda Magalhães da Silva, Girliani Silva de Sousa, Indara Cavalcante Bezerra, Christina César Praça Brasil, Luiza Jane Eyre de Souza Vieira, Fernanda Colares de Borba Netto, José Maria Ximenes Guimarães, Maria Cecilia de Souza Minayo.

**Writing – review & editing:** Jonas Loiola Gonçalves, Raimunda Magalhães da Silva, Girliani Silva de Sousa, Indara Cavalcante Bezerra, Christina César Praça Brasil, Luiza Jane Eyre de Souza Vieira, Fernanda Colares de Borba Netto, José Maria Ximenes Guimarães, Maria Cecilia de Souza Minayo.

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
