## [Decision Letter · Decision Letter 0]

4 Oct 2023

PONE-D-23-17539Dependent older adults’ care barriers: brazilian primary health care managers’ perspectivePLOS ONE

Dear Dr. Gonçalves,

Thank you for submitting your manuscript to PLOS ONE. After careful consideration, we feel that it has merit but does not fully meet PLOS ONE’s publication criteria as it currently stands. Therefore, we invite you to submit a revised version of the manuscript that addresses the points raised during the review process.

We look forward to receiving your revised manuscript.

Kind regards,

Marília Jesus Batista de Brito Mota, Post-doc

Academic Editor

PLOS ONE

Journal Requirements:

Additional Editor Comments :

The manuscript is very relevant considering the aging rates observed around the world. The study provides data to guide public policies and needs to be revised in some aspects described by reviewer 1.

Reviewers' comments:

Reviewer's Responses to Questions

**Comments to the Author**

1. Is the manuscript technically sound, and do the data support the conclusions?

Reviewer #1: Yes

2. Has the statistical analysis been performed appropriately and rigorously? 

Reviewer #1: N/A

3. Have the authors made all data underlying the findings in their manuscript fully available?

Reviewer #1: Yes

4. Is the manuscript presented in an intelligible fashion and written in standard English?

Reviewer #1: Yes

5. Review Comments to the Author

Reviewer #1: Current and relevant topic in view of the socio-sanitary impacts of the demographic transition. The text analyzes important issues for the formulation of public policies, as well as the organization of care practices for the elderly, especially for the elderly, that is, those over 80 years of age.

The article is theoretically and methodologically structured, with clear arguments and relevant dialogues with national and international publications on the subject.

Despite these considerations on the merits of the article, I highlight some aspects that deserve necessary revisions for publication:

I. Introduction

1. The Brazilian Institute of Geography and Statistics (IBGE) published the 2022 Demographic Census. I suggest updating the demographic data presented in the article.

2. The results presented are part of a larger research, as the authors mention (lines 126-131). However, the choice of the objective of this article with other aspects found is not sufficiently clarified. It inappropriate to say “the entire collection of this knowledge is registered in a thematic issue of the Ciência & Saúde Coletiva jornal” (lines 160-161). There are in this publication, for example, other articles that do not deal with the same theme.

3. Programa Acompanhante de Idosos (PAI) of the Municipal Secretary of Health of São Paulo is the correct name. It’s necessary to change (line 82) “São Paulo has the Acompanhamento ao Idoso (“Elderly Monitoring”)...”.

II. Methods

1. Presents theoretical foundation, description of procedures and ethical aspects of the research. However, It’s appropriate to rephrase the paragraph (lines 138-143) “... all of them (including Araranguá) share that they are hubs for gathering the demand of SUS users since they have all the equipment and resources for primary, secondary, tertiary, and ultra-specialized care” The reference (22) is insufficient to characterize the health care network. Publications in reference (23) have more adequate paragraphs about it.

2. 16 study participants in the 8 cities. It’s necessary to describe this distribution, as also the responsibilities inherent in managing care for dependent older adults.

3. It’s advisable to specify the software used in the research (line 172).

III. Results

It’s appropriate to clarify acronyms and citation some excerpts from the interviews – “PAIF”, “Mais Médicos”, “PNAB”.

Note: also in other parts of the article, such as “PHC”, “SUS”.

IV. Discussion

In general, this part of the text is well founded with current publications, supported by research in the Brazilian reality, as well as international experiences, enabling comparative analyzes and possible contributions to the Brazilian context.

However, It’s necessary to review some aspects:

1. The dimension “barriers caused by problems in family care for the dependent older adults” has not been sufficiently analysed.

2. There are participants who have been worked in Belo Horizonte. However They didn’t talk about ”Greater Care”. Fact that deserves discussion.

3. The argument in lines 386-394 doesn’t establish a dialogue necessary with lack of knowledge of the subjects of the policy.

4. Including in the Introduction the paragraph (lines 402-409) is more adequate.

5. It’s necessary discuss more appropriately with the results the argument “It is a consensus among the authors that a health hospital-and-medical professional-centered system alone will have to be modified” (lines 450-451).

6. PLOS authors have the option to publish the peer review history of their article (what does this mean?). If published, this will include your full peer review and any attached files.

Reviewer #1: No

---

## [Author Response · Author response to Decision Letter 0]

21 Nov 2023

Thank you for your contributions to our material.

We proceed with the adjustments as requested.

Regarding the presentation letter, we complement it with:

It was submitted and approved by the Research Ethics Committee of the Oswaldo Cruz Foundation/Rio de Janeiro, through the Certificate of Presentation of Ethical Review (44615315.0.0000.5240), sound of number 1.326.631. We declare that the data of this study are supplementary to the submission of the manuscript, given the ethical restrictions of the Brazilian legislation with human beings.

We declare and thank the Ceará Foundation for the Support of Scientific and Technological Development for granting a master's scholarship in Collective Health to the student Jonas Loiola Gonçalves.

The authors received no specific funding for this work.

We also complement the submission with supplementary material on data availability.

Regarding the required revisions, we sequence them as follows:

Introduction

1. The Brazilian Institute of Geography and Statistics (IBGE) published the 2022 Demographic Census. I suggest updating the demographic data presented in the article.

Response: Population aging is almost widespread globally, albeit in heterogeneous epidemiological rhythms. The 2023 Brazilian Census reveals that the elderly population has reached 31,2 million, representing 14,7% of the brazilian, the growth of this population represented 39.8% from 2012 to 2021¹.

2. The results presented are part of a larger research, as the authors mention (lines 126-131). However, the choice of the objective of this article with other aspects found is not sufficiently clarified. It inappropriate to say “the entire collection of this knowledge is registered in a thematic issue of the Ciência & Saúde Coletiva jornal” (lines 160-161). There are in this publication, for example, other articles that do not deal with the same theme.

Response: We present a preview of the results in a collection registered in a thematic issue of the Ciência & Saúde Coletiva journal23. The managers, the main stakeholders in this study were heard through a semi-structured interview at a pre-established date, time, and place.

3. Programa Acompanhante de Idosos (PAI) of the Municipal Secretary of Health of São Paulo is the correct name. It’s necessary to change (line 82) “São Paulo has the Acompanhamento ao Idoso (“Elderly Monitoring”)...”.

Response: São Paulo has the “Programa Acompanhamente de Idosos” (PAI) (“Elderly Monitoring”) Program, which provides a multidisciplinary team to support the most fragile and dependent, including meeting their needs of daily life.

Methods

1. Presents theoretical foundation, description of procedures and ethical aspects of the research. However, It’s appropriate to rephrase the paragraph (lines 138-143) “... all of them (including Araranguá) share that they are hubs for gathering the demand of SUS users since they have all the equipment and resources for primary, secondary, tertiary, and ultra-specialized care” The reference (22) is insufficient to characterize the health care network. Publications in reference (23) have more adequate paragraphs aboutit.

Response: However, all of them (including Araranguá) are hub regions for meeting the demands of the Unified Health System (SUS), providing users with the equipment and resources necessary for care in the areas of primary, secondary, tertiary, and ultra-specialized care22-23.

2. 16 study participants in the 8 cities. It’s necessary to describe this distribution, as also the responsibilities inherent in managing care for dependent older adults.

Response: Study participants form a sample of 16 managers from five different country regions, distribution in: Araranguá – SC (4) , Brasília – DF (1), Fortaleza-CE (2), Manaus – AM (2), Porto Alegre – RS (1), Belo Horizonte – MG (4), Rio de Janeiro - RJ (1) e Teresina – PI (1). They are also selected by convenience and because they are firmly integrated with the local administration and with responsibilities inherent in managing care for dependent older adults, managers who coordinate and are responsible for managing basic care units, programs and specialized services offering health care to dependent elderly people.

3. It’s advisable to specify the software used in the research (line 172).

Response: The statements were fully transcribed and organized manually, without software support.

III. Results

It’s appropriate to clarify acronyms and citation some excerpts from the interviews – “PAIF”, “Mais Médicos”, “PNAB”. Note: also in other parts of the article, such as “PHC”, “SUS”.

Response: São Paulo has the “Programa Acompanhamente de Idosos” (PAI) (“Elderly Monitoring”)

In Rio de Janeiro, primary care offers care to bedridden older adults. However, the health workers and the Primary Health Care (PHC) ’ team recognize they can do very little and cannot attend to all older adults in need6.

Structuring an Elderly Care Network regulated by Primary Care requires close communication between the Unified Health System (SUS) and the Unified Social Assistance System (SUAS).

The team already supports other things. So, on this health issue, the Family Protection and Comprehensive Care Service (PAIF) team is very overloaded. This situation brings uncertainty about how to proceed with this professional. (Manager 3 – Belo Horizonte)

Our units are overloaded. We have several inhabitants well above what the National basic care policy (PNAB) recommends for each strategy. So, this hinders providing more detailed care for older adults. (Manager 4 – Araranguá)

The “Programa Mais Médicos” (Program More Doctors)

IV. Discussion

The dimension “barriers caused by problems in family care for the dependent older adults” has not been sufficiently analysed.

Response: 

Family barriers that permeate the accelerated growth of the older population contradict the low birth rates, as well as changes in marriage rates and the insertion of women in the job market. It is evident that Brazilian women are increasingly assuming the role of provider, but still maintain responsibility for the care of those who are dependent, changes that directly affect the ability of families to guarantee and offer care to the elderly population27.

Furthermore, the work of these family caregivers is often uninterrupted and solitary, without the support of health services and public protection policies. There is a need for public support policies for dependent elderly people and their caregivers, with the aim of overcoming barriers and subsidizing better care for everyone27-28.

There are participants who have been worked in Belo Horizonte. However They didn’t talk about ”Greater Care”. Fact that deserves discussion.

Response: We complement the interpretation, of this request.

The argument in lines 386-394 doesn’t establish a dialogue necessary with lack of knowledge of the subjects of the policy.

The Policy document17 mentions dependent older adults in two stages: in the introduction and the chapter addressing the integrality of the actions, highlighting a concern when managers are unaware of the policy issues, especially regarding the themes of dependent elderly people. Because the oldest with functional difficulties stood at 2%-4% at the time of writing. That may be why the emphasis was on healthy aging, and the loss of autonomy deserved only general considerations. In the 2013 National Health Survey, they found 8.4% of older adults dependent on others to perform the activities of daily living, and 22.0% for instrumental activities, showing that the percentage of older adults who are losing autonomy increases while the number of those aged 80 and over is growing rapidly30.

It’s necessary discuss more appropriately with the results the argument “It is a consensus among the authors that a health hospital-and-medical professional-centered system alone will have to be modified” (lines 450-451).

Response:

It is a consensus among the authors that a health system centered on the hospital and the medical professional is not sufficient to meet the demands of dependent elderly people, there is a need for training, public policies, and dialogue with the whole of society. Investments in Primary Care to reduce and consolidate care are necessary in an elderly society whose future is already present, including in low- and middle-income countries. The authors highlight that Japan has the most consolidated primary care services. The Japanese government implemented the “Community-Based Integrated Care System” to offer practical support to people at the end of life, guided by four elements: self-help, mutual help, social solidarity care and government care36.

---

## [Decision Letter · Decision Letter 1]

7 Feb 2024

PONE-D-23-17539R1Dependent older adults’ care barriers: brazilian primary health care managers’ perspectivePLOS ONE

Dear Dr. Gonçalves,

Thank you for submitting your manuscript to PLOS ONE. After careful consideration, we feel that it has merit but does not fully meet PLOS ONE’s publication criteria as it currently stands. Therefore, we invite you to submit a revised version of the manuscript that addresses the points raised during the review process.

We look forward to receiving your revised manuscript.

Kind regards,

Marília Jesus Batista de Brito Mota, Post-doc

Academic Editor

PLOS ONE

Additional Editor Comments:

"Dependent older adults’ care barriers: brazilian primary health care managers’ perspective"

The authors have answered the points in the first review. The manuscript brings a very relevant issue about the barriers for atend dependent older adults in Primary Health Care. The problem of the manuscript is that the results is focused in national context and would be more appropriated national publication. Many importante points were evaluated for the new reviewers in this fase of the editorial process in order to improve the manuscript to be eligible for publication. After the changes the can submit again the manuscript.

Reviewer 1

Introduction:

1- It is bureaucratic and, at times, generic. It needs to be very clear about what is meant by "dependent adult". It needs to explain to the foreign reader in a synthetic way how PHC is organized in Brazil and how it is articulated in the elderly care network, etc. Make it clearer what it is, including SUS!!!

2- References are missing in paragraphs 4, 6, 7, 8, 9 and many others.

3- “In Brazil, PHC’s attributions are promoting health, preventing diseases, and providing care7”. In this sentence, you need to be careful, since it cannot be just in Brazil, as they are attributes of (non-selective) APS internationally.

4- In the sentence “Structuring an Elderly Care Network regulated by Primary Care requires close communication between the Unified Health System (SUS) and the Unified Social Assistance System (SUAS), as recommended in other countries”, a reference is needed at the end of the sentence and, Also, justify why? It was confusing and also does not make it clear to the non-Brazilian reader what SUS and SUAS mean.

5- In general, the introduction paragraphs are not articulated and are only descriptive, lacking some criticality in the writing and approach. The problem is not well discussed in the introduction. According to the research question, discuss PHC and its relationship with elderly care in Brazil and around the world. The question is not thematized in the introduction, which raises relevant questions, although disjointed and without debate appropriate to the problem presented. The barriers were not discussed in the text or were poorly explained in the description.

Materials and methods:

6- There are many references missing in the paragraphs to strengthen the argument.

7- Where and how was it used in hermeneutic-dialectic analysis? There is nothing to demonstrate its use. In what aspects did HD help in understanding the results and their discussion?

8- Is HD possible without data triangulation? Why does this research, the result of a larger and richer research as a whole, boil down to treating the perception of managers without triangulating it with other data, including sociodemographic data?

9- Why and how were the cities chosen? Why is one of them not a state capital and what is the implication of this in the analyses?

10- Why 4 managers in Araranguá and fewer in the others?

11- I suggest a map with the location of these municipalities in the country for better visualization.

12- Also, a table is needed with data from these municipalities: population, PHC coverage, other services, number of elderly people, epidemiological data, etc.

13- The authors state that the choice was made for convenience, but what is appropriate in qualitative research is an intentional choice!!! (could you justify it better?).

14- Regarding the interviews, you need to provide details with COREQ and not just indicate what you used, as important details are missing (Who carried them out? See Quoreq and try to bring up aspects that clarify the conditions of data production). The average interview time was 30 minutes? (Only? What depth is possible in such short interviews?). Explain in the method how you arrived at the categories/topics.

Results:

15- The fragments are interesting and rich. However, there are two general problems: the description of the results is VERY synthetic and does not explore the wealth of information. In this sense, the description is limited to repeating the managers' arguments as if the statements contained a truth in themselves;

16- As a result, the text contains many fragments (empirical results), but does not explore them and, consequently, leaves it up to the reader to interpret the results. In my understanding, this choice impoverished many of the data found and compromised the entire article, making it fragile and without sufficient argumentative strength.

17- There would be a need to rewrite all the results, even if the chosen topics were maintained, but described with more attention and more criticality (the description does NOT mean repetition of arguments, but elaboration of text that synthesizes the ideas contained explicitly or implicitly in the managers' fragments).

18- Are the categories/topics common for all elected municipalities? Is there no diversity among managers? Wouldn't this be an aspect worth considering?

Discussion:

19- I feel that the discussion does not fulfill the promise of hermeneutic-dialectic analysis. Or I would need to demonstrate the strength of dialectics in writing. The text is quite descriptive and, to some extent, repeats the results without sufficient criticality, historicity or dialectics.

20- In the way they judge families, there would be a need to consider the different family contexts across the country (demographic and socioeconomic data added to inferences that consider national complexity in different regions of the country).

21- Are there no differences between regions and cities? How can a “poor” Brazilian family take care of a dependent elderly person? How do different family arrangements manage to care for the elderly? As it stands, there is a homogeneity in the way of thinking about such families' responsibilities and viability.

22- Managers point out in a bureaucratic way, but the analysis needs to counter argue the findings.

It is not understandable because methodologically different cities/regions of the country were used and this diversity does not appear in the results or in the discussion. It needs to be justified or at least revised. I think this impoverished and compromised the quality and originality of the article, as everything became generic.

The discussion with the literature is bureaucratic and prescriptive without any argumentative force that goes beyond the description and compilation of national and international findings. I think there is a need for rewriting and new arguments to make the text rich and capable of providing aspects that contribute to research on elderly care.

Finally, the discussion does NOT answer the research question sufficiently, as APS practically disappears from the analyses. I think that the authors would need to review the entire analytical approach.

Final considerations: insufficient and generic. It needs to answer the research question supported by the results. But bring perspectives of intervention in politics in a less generic way.

Reviewer 2

The text discuss a very relevant issue in Brazilian reality and has an interesting focus to present the perceptions of the managers of health services towards barriers in care for older adults.

The text is confusing sometimes due to the use of prepositions that may not be the most appropriate, literal translation of expressions or idioms that may not have the same meaning in English, and other aspects that require the support of a professional reviewer that may have access to the text written in the Portuguese for the correct alterations.

As for the structure of the text, in the results section we advise the authors to check how the statements presented may reinforce or build upon what has been presented and will be presented in the discussion section. At one point the authors suggest a comparison between Brazil's health system and other countries, but forget to explain that Brazilian's federalism, consisted of 3 federative beings, is not easily comparable due to its particularities. This makes the implementation of primary health a particular challenge in Brazil. There is a lack of connection of primary care for dependent older adults and the Brazilian family health strategy, this could be an important aspect of the implementation of services.

In discussion, the authors present a figure related to older adults who are responsible for families (line 400), but present an outdated figure, the same author has much recent work on the subject, even including the effect of the Covid-19 outbreak on families and the income of older populations. Lines 419-421 repeat the figures in the introduction, which are not from the "Censo" but from another household research, make sure you cite the updated values that increase 60+ participation in our society. Content between lines 441-443 were not very clear, and need careful review. Line 477 cites a study by Brazilian authors that is not presented in the reference list. The last paragraphs before conclusions cite Japanese examples and possible adaptations but does not suggest how to adapt and which services could be formulated for dependent older adults.

There could be a paragraph where some of the limitations of the study are presented (Why 4 participants in a middle-size city and just 1 in most of the largest cities? How can this short list of professionals limit or influence the results? How was the interview carried and what were some of the possible limitations in the interview script? and other things that they may consider relevant).

We hope our suggestion can improve and better connect what has been presented throughout the article with the relevant debate of the subject.

Reviewers' comments:

Reviewer's Responses to Questions

**Comments to the Author**

1. If the authors have adequately addressed your comments raised in a previous round of review and you feel that this manuscript is now acceptable for publication, you may indicate that here to bypass the “Comments to the Author” section, enter your conflict of interest statement in the “Confidential to Editor” section, and submit your "Accept" recommendation.

Reviewer #2: All comments have been addressed

Reviewer #3: (No Response)

2. Is the manuscript technically sound, and do the data support the conclusions?

Reviewer #2: Partly

Reviewer #3: Partly

3. Has the statistical analysis been performed appropriately and rigorously? 

Reviewer #2: N/A

Reviewer #3: N/A

4. Have the authors made all data underlying the findings in their manuscript fully available?

Reviewer #2: No

Reviewer #3: Yes

5. Is the manuscript presented in an intelligible fashion and written in standard English?

Reviewer #2: Yes

Reviewer #3: Yes

6. Review Comments to the Author

Reviewer #2: Introduction: It is bureaucratic and, at times, generic. It needs to be very clear about what is meant by "dependent adult". It needs to explain to the foreign reader in a synthetic way how PHC is organized in Brazil and how it is articulated in the elderly care network, etc. Make it clearer what it is, including SUS!!! References are missing in paragraphs 4, 6, 7, 8, 9 and many others. “In Brazil, PHC’s attributions are promoting health, preventing diseases, and providing care7”. In this sentence, you need to be careful, since it cannot be just in Brazil, as they are attributes of (non-selective) APS internationally. In the sentence “Structuring an Elderly Care Network regulated by Primary Care requires close communication between the Unified Health System (SUS) and the Unified Social Assistance System (SUAS), as recommended in other countries”, a reference is needed at the end of the sentence and, Also, justify why? It was confusing and also does not make it clear to the non-Brazilian reader what SUS and SUAS mean. In general, the introduction paragraphs are not articulated and are only descriptive, lacking some criticality in the writing and approach. The problem is not well discussed in the introduction. According to the research question, discuss PHC and its relationship with elderly care in Brazil and around the world. The question is not thematized in the introduction, which raises relevant questions, although disjointed and without debate appropriate to the problem presented. The barriers were not discussed in the text or were poorly explained in the description.

Materials and methods: There are many references missing in the paragraphs to strengthen the argument. Where and how was it used in hermeneutic-dialectic analysis? There is nothing to demonstrate its use. In what aspects did HD help in understanding the results and their discussion? Is HD possible without data triangulation? Why does this research, the result of a larger and richer research as a whole, boil down to treating the perception of managers without triangulating it with other data, including sociodemographic data? Why and how were the cities chosen? Why is one of them not a state capital and what is the implication of this in the analyses? Why 4 managers in Araranguá and fewer in the others? I suggest a map with the location of these municipalities in the country for better visualization. Also, a table is needed with data from these municipalities: population, PHC coverage, other services, number of elderly people, epidemiological data, etc. The authors state that the choice was made for convenience, but what is appropriate in qualitative research is an intentional choice!!! (could you justify it better?). Regarding the interviews, you need to provide details with COREQ and not just indicate what you used, as important details are missing (Who carried them out? See Quoreq and try to bring up aspects that clarify the conditions of data production). The average interview time was 30 minutes? (Only? What depth is possible in such short interviews?). Explain in the method how you arrived at the categories/topics.

Results: The fragments are interesting and rich. However, there are two general problems: the description of the results is VERY synthetic and does not explore the wealth of information. In this sense, the description is limited to repeating the managers' arguments as if the statements contained a truth in themselves; 2) As a result, the text contains many fragments (empirical results), but does not explore them and, consequently, leaves it up to the reader to interpret the results. In my understanding, this choice impoverished many of the data found and compromised the entire article, making it fragile and without sufficient argumentative strength. There would be a need to rewrite all the results, even if the chosen topics were maintained, but described with more attention and more criticality (the description does NOT mean repetition of arguments, but elaboration of text that synthesizes the ideas contained explicitly or implicitly in the managers' fragments). Are the categories/topics common for all elected municipalities? Is there no diversity among managers? Wouldn't this be an aspect worth considering?

Discussion: I feel that the discussion does not fulfill the promise of hermeneutic-dialectic analysis. Or I would need to demonstrate the strength of dialectics in writing. The text is quite descriptive and, to some extent, repeats the results without sufficient criticality, historicity or dialectics. In the way they judge families, there would be a need to consider the different family contexts across the country (demographic and socioeconomic data added to inferences that consider national complexity in different regions of the country). Are there no differences between regions and cities? How can a “poor” Brazilian family take care of a dependent elderly person? How do different family arrangements manage to care for the elderly? As it stands, there is a homogeneity in the way of thinking about such families' responsibilities and viability. Managers point out in a bureaucratic way, but the analysis needs to counter argue the findings.

It is not understandable because methodologically different cities/regions of the country were used and this diversity does not appear in the results or in the discussion. It needs to be justified or at least revised. I think this impoverished and compromised the quality and originality of the article, as everything became generic.

The discussion with the literature is bureaucratic and prescriptive without any argumentative force that goes beyond the description and compilation of national and international findings. I think there is a need for rewriting and new arguments to make the text rich and capable of providing aspects that contribute to research on elderly care.

Finally, the discussion does NOT answer the research question sufficiently, as APS practically disappears from the analyses. I think that the authors would need to review the entire analytical approach.

Final considerations: insufficient and generic. It needs to answer the research question supported by the results. But bring perspectives of intervention in politics in a less generic way.

Reviewer #3: The text discuss a very relevant issue in Brazilian reality and has an interesting focus to present the perceptions of the managers of health services towards barriers in care for older adults.

The text is confusing sometimes due to the use of prepositions that may not be the most appropriate, literal translation of expressions or idioms that may not have the same meaning in English, and other aspects that require the support of a professional reviewer that may have access to the text written in the Portuguese for the correct alterations.

As for the structure of the text, in the results section we advise the authors to check how the statements presented may reinforce or build upon what has been presented and will be presented in the discussion section. At one point the authors suggest a comparison between Brazil's health system and other countries, but forget to explain that Brazilian's federalism, consisted of 3 federative beings, is not easily comparable due to its particularities. This makes the implementation of primary health a particular challenge in Brazil. There is a lack of connection of primary care for dependent older adults and the Brazilian family health strategy, this could be an important aspect of the implementation of services.

In discussion, the authors present a figure related to older adults who are responsible for families (line 400), but present an outdated figure, the same author has much recent work on the subject, even including the effect of the Covid-19 outbreak on families and the income of older populations. Lines 419-421 repeat the figures in the introduction, which are not from the "Censo" but from another household research, make sure you cite the updated values that increase 60+ participation in our society. Content between lines 441-443 were not very clear, and need careful review. Line 477 cites a study by Brazilian authors that is not presented in the reference list. The last paragraphs before conclusions cite Japanese examples and possible adaptations but does not suggest how to adapt and which services could be formulated for dependent older adults.

There could be a paragraph where some of the limitations of the study are presented (Why 4 participants in a middle-size city and just 1 in most of the largest cities? How can this short list of professionals limit or influence the results? How was the interview carried and what were some of the possible limitations in the interview script? and other things that they may consider relevant).

We hope our suggestion can improve and better connect what has been presented throughout the article with the relevant debate of the subject.

7. PLOS authors have the option to publish the peer review history of their article (what does this mean?). If published, this will include your full peer review and any attached files.

Reviewer #2: No

Reviewer #3: No

---

## [Author Response · Author response to Decision Letter 1]

23 Mar 2024

REVIEWER 1 

Introduction:

1- It is bureaucratic and, at times, generic. It needs to be very clear about what is meant by "dependent adult". It needs to explain to the foreign reader in a synthetic way how PHC is organized in Brazil and how it is articulated in the elderly care network, etc. Make it clearer what it is, including SUS!!!

R. We seek to meet this suggestion and leave the text as follows: It is important to note that in the Brazilian situation, before the promulgation of the Federal Constitution, the elderly population suffered from incipient and charitable government actions. From its institution, the SUS was created, a health instance focused on the universal right to health. An articulation with the Unified Social Assistance System (SUAS) emerged, in the search to overcome difficulties and guarantee social protection for the population, based on support for people, families and their community contexts.

2- References are missing in paragraphs 4, 6, 7, 8, 9 and many others.

misconception:

R. Reviewed and presented; we apologize for this misconception presented in review 1.

3- “In Brazil, PHC’s attributions are promoting health, preventing diseases, and providing care7”. In this sentence, you need to be careful, since it cannot be just in Brazil, as they are attributes of (non-selective) APS internationally.

R. Based on the suggestion, we adapted it to the following text: In the case Brazil and other international contexts, the attributions of PHC’s (non-seletive) are promoting health, preventing diseases, and providing care.

4- In the sentence “Structuring an Elderly Care Network regulated by Primary Care requires close communication between the Unified Health System (SUS) and the Unified Social Assistance System (SUAS), as recommended in other countries”, a reference is needed at the end of the sentence and, Also, justify why? It was confusing and also does not make it clear to the non-Brazilian reader what SUS and SUAS mean.

We seek to meet this suggestion and leave the text as follows: It is important to note that in the Brazilian situation, before the promulgation of the Federal Constitution, the elderly population suffered from incipient and charitable government actions. From its institution, the SUS was created, a health instance focused on the universal right to health. An articulation with the Unified Social Assistance System (SUAS) emerged, in the search to overcome difficulties and guarantee social protection for the population, based on support for people, families and their community contexts.

5- In general, the introduction paragraphs are not articulated and are only descriptive, lacking some criticality in the writing and approach. The problem is not well discussed in the introduction. According to the research question, discuss PHC and its relationship with elderly care in Brazil and around the world. The question is not thematized in the introduction, which raises relevant questions, although disjointed and without debate appropriate to the problem presented. The barriers were not discussed in the text or were poorly explained in the description.

R. Based on the reordering and adjustments suggested above, we sought to have taken into account the suggestions of reviewer 1, taking into account this comment.

Materials and methods:

6- There are many references missing in the paragraphs to strengthen the argument.

R. Reviewed and presented; we apologize for this misconception presented in review 1.

7- Where and how was it used in hermeneutic-dialectic analysis? There is nothing to demonstrate its use. In what aspects did HD help in understanding the results and their discussion?

R: We complement the material with the following paragraphs, seeking to meet the present suggestion:

We developed a qualitative study anchored in the hermeneutic theoretical framework that is based on the understanding of social processes, based on the communication of daily life, common sense, considering that the human being is permeated by its historicity, and the interconnections come from social and cultural contexts20-21. 

Based on the experiences, we tried to analyze a consensus of the experiences, senses, meanings and symbols contained in the participants' testimonies. Thus, a comprehensive reflection emerges on the barriers pointed out by PHC health managers regarding the adequate provision of health care to dependent elderly people living at home20-21.

8- Is HD possible without data triangulation? Why does this research, the result of a larger and richer research as a whole, boil down to treating the perception of managers without triangulating it with other data, including sociodemographic data?

R. We introduce and justify the triagulation as presented below: The study is guided by the assumptions of triangulation, considering of the participation of places, people, researchers, theories, and the intra-method20. Involving the understanding of the work of PHC (health workers, professionals who work in the units and managers) in the face of the interlocutions of the situation of the elderly, their families and family caregivers6.

9- Why and how were the cities chosen? Why is one of them not a state capital and what is the implication of this in the analyses? 10- Why 4 managers in Araranguá and fewer in the others?

R. View limitations at study inserted conform review 2.

11- I suggest a map with the location of these municipalities in the country for better visualization.

R. Map presented from the inclusion of figure 1, as requested.

12- Also, a table is needed with data from these municipalities: population, PHC coverage, other services, number of elderly people, epidemiological data, etc.

R. We seek to meet this suggestion without losing the essence of qualitative research, since its currents are not based on positivist assumptions.

13- The authors state that the choice was made for convenience, but what is appropriate in qualitative research is an intentional choice!!! (could you justify it better?).

 R. We follow the suggestion and present the following text: Selected by intentionality and because they are firmly integrated with the local administration and with responsibilities inherent in managing care for dependent older adults, managers who coordinate and are responsible for managing basic care units, programs and specialized services offering health care to dependent elderly people.

14- Regarding the interviews, you need to provide details with COREQ and not just indicate what you used, as important details are missing (Who carried them out? See Quoreq and try to bring up aspects that clarify the conditions of data production). The average interview time was 30 minutes? (Only? What depth is possible in such short interviews?). Explain in the method how you arrived at the categories/topics.

R. This suggestion was heeded and consolidated throughout the manuscript.

Results

15- The fragments are interesting and rich. However, there are two general problems: the description of the results is VERY synthetic and does not explore the wealth of information. In this sense, the description is limited to repeating the managers' arguments as if the statements contained a truth in themselves; 16- As a result, the text contains many fragments (empirical results), but does not explore them and, consequently, leaves it up to the reader to interpret the results. In my understanding, this choice impoverished many of the data found and compromised the entire article, making it fragile and without sufficient argumentative strength. 17- There would be a need to rewrite all the results, even if the chosen topics were maintained, but described with more attention and more criticality (the description does NOT mean repetition of arguments, but elaboration of text that synthesizes the ideas contained explicitly or implicitly in the managers' fragments). 18- Are the categories/topics common for all elected municipalities? Is there no diversity among managers? Wouldn't this be an aspect worth considering?

R. Based on the suggestions of both reviewers, we seek to respond promptly to this request, and we hope that our efforts have been able to achieve this request.

Discussion

19- I feel that the discussion does not fulfill the promise of hermeneutic-dialectic analysis. Or I would need to demonstrate the strength of dialectics in writing. The text is quite descriptive and, to some extent, repeats the results without sufficient criticality, historicity or dialectics.

R. We have adopted the hermenutic path based on the suggestions in the methods, we have revisited all the material, especially on this topic, so we hope to have responded to this suggestion.

20- In the way they judge families, there would be a need to consider the different family contexts across the country (demographic and socioeconomic data added to inferences that consider national complexity in different regions of the country).

R. We seek to bring the social data presented since the first version, in which in this second review we updated indicators and the introduction of the issue of Brazilian federalism as presented throughout the manuscript.

21- Are there no differences between regions and cities? How can a “poor” Brazilian family take care of a dependent elderly person? How do different family arrangements manage to care for the elderly? As it stands, there is a homogeneity in the way of thinking about such families' responsibilities and viability. 22- Managers point out in a bureaucratic way, but the analysis needs to counter argue the findings.

R. Based on the previous suggestions, we hope we have answered this question.

Final considerations 

Insufficient and generic. It needs to answer the research question supported by the results. But bring perspectives of intervention in politics in a less generic way.

We thank you for the suggestion of redesigning the final considerations to remedy the generic and we present this request in the base text.

Reviewer 2

The text is confusing sometimes due to the use of prepositions that may not be the most appropriate, literal translation of expressions or idioms that may not have the same meaning in English, and other aspects that require the support of a professional reviewer that may have access to the text written in the Portuguese for the correct alterations.

R. We develop the spelling review of the material with a specialized professional as suggested by the reviewer 2.

As for the structure of the text, in the results section we advise the authors to check how the statements presented may reinforce or build upon what has been presented and will be presented in the discussion section. 

R. We verified the empirical material, seeking to reinforce the analysis and discussion, presented throughout the sections as requested.

At one point the authors suggest a comparison between Brazil's health system and other countries, but forget to explain that Brazilian's federalism, consisted of 3 federative beings, is not easily comparable due to its particularities. This makes the implementation of primary health a particular challenge in Brazil. There is a lack of connection of primary care for dependent older adults and the Brazilian family health strategy, this could be an important aspect of the implementation of services.

R. We complement the review questioning, which is presented as follows: Stands out n the Brazilian reality, federalism, composed of three federative entities, is not easily comparable due to its particularities, making the implementation of PHC a particular challenge in Brazil. The disarticulation of PHC and the family health strategy in the care of the dependent elderly suffers from barriers to the implementation of primary services in a universal way.

In discussion, the authors present a figure related to older adults who are responsible for families (line 400), but present an outdated figure, the same author has much recent work on the subject, even including the effect of the Covid-19 outbreak on families and the income of older populations.

R. We have made the necessary update in the text as well as at the end (references): A paradox is currently experienced by families with older adults in their homes. More than 90,1% of these people help maintain or maintain families alone.

Lines 419-421 repeat the figures in the introduction, which are not from the "Censo" but from another household research, make sure you cite the updated values that increase 60+ participation in our society. 

R. We chose to promote these data only in the introductory framework, improving the discursive synthesis here.

Content between lines 441-443 were not very clear, and need careful review. Line 477 cites a study by Brazilian authors that is not presented in the reference list. 

R. Adjustment made, based on the support material presented. It was without the proper reference.

The last paragraphs before conclusions cite Japanese examples and possible adaptations but does not suggest how to adapt and which services could be formulated for dependent older adults.

R. Reformulation promptly complied with, with base text presented: An example that could be adapted to the Brazilian reality, based on the implementation of long-term care, institutional investments that meet the needs of health managers for the proper consolidation of PHC and the family health strategy. Thus, strengthening intersectoral relationships centered on home care. Since it is necessary to take better care of older adults in Brazil and the world that is aging36-37.

There could be a paragraph where some of the limitations of the study are presented (Why 4 participants in a middle-size city and just 1 in most of the largest cities? How can this short list of professionals limit or influence the results? How was the interview carried and what were some of the possible limitations in the interview script? and other things that they may consider relevant).

R. We have inserted this important paragraph as presented: The limitations of the study include the unavailability of carrying out focus groups between managers and federative units. Another difficulty is highlighted by access to data collection, given the time that management has for the interview, limited physical spaces and contexts of urban violence that directly implicate research processes in Brazil.

We hope our suggestion can improve and better connect what has been presented throughout the article with the relevant debate of the subject

R. We greatly appreciate the reviewer's contributions in fostering valuable improvements to the subject and theme.

---

## [Decision Letter · Decision Letter 2]

9 Aug 2024

Barriers to care for dependent older adults: Brazilian Primary Health Care managers’ perspective

PONE-D-23-17539R2

Dear Dr. Gonçalves,

We’re pleased to inform you that your manuscript has been judged scientifically suitable for publication and will be formally accepted for publication once it meets all outstanding technical requirements.

Kind regards,

Marília Jesus Batista de Brito Mota, Post-doc

Academic Editor

PLOS ONE

Additional Editor Comments (optional):

I would like to explain in advance and apologize for the delay in the editorial process. This was due to the difficulty in finding reviewers for this manuscript, but we were able to complete the process satisfactorily.

The reviewers of this manuscript found that the authors had made all the recommended changes. The decision is therefore to accept the manuscript for publication.

Reviewers' comments:

Reviewer's Responses to Questions

**Comments to the Author**

1. If the authors have adequately addressed your comments raised in a previous round of review and you feel that this manuscript is now acceptable for publication, you may indicate that here to bypass the “Comments to the Author” section, enter your conflict of interest statement in the “Confidential to Editor” section, and submit your "Accept" recommendation.

Reviewer #1: All comments have been addressed

Reviewer #3: All comments have been addressed

2. Is the manuscript technically sound, and do the data support the conclusions?

Reviewer #1: Yes

Reviewer #3: Yes

3. Has the statistical analysis been performed appropriately and rigorously? 

Reviewer #1: N/A

Reviewer #3: N/A

4. Have the authors made all data underlying the findings in their manuscript fully available?

Reviewer #1: Yes

Reviewer #3: No

5. Is the manuscript presented in an intelligible fashion and written in standard English?

Reviewer #1: Yes

Reviewer #3: Yes

6. Review Comments to the Author

Reviewer #1: (No Response)

Reviewer #3: The authors have taken reviews into consideration and improved the manuscript. From my perspective the text discuss a relevant issue and bring views from different locations. Maybe more contradictions in terms of PHC promotion, care and access could have been presented, however the text achieves the objective aimed.

7. PLOS authors have the option to publish the peer review history of their article (what does this mean?). If published, this will include your full peer review and any attached files.

Reviewer #1: No

Reviewer #3: No

---

## [Editor Report · Acceptance letter]

15 Aug 2024

PONE-D-23-17539R2 

PLOS ONE

Dear Dr. Gonçalves, 

I'm pleased to inform you that your manuscript has been deemed suitable for publication in PLOS ONE. Congratulations! Your manuscript is now being handed over to our production team.

Kind regards, 

on behalf of

Professor Marília Jesus Batista de Brito Mota 

Academic Editor

PLOS ONE